# Attitude toward vaccination against COVID-19 and acceptance of the national "QazVac" vaccine in the Aktobe city population, West Kazakhstan: A cross-sectional survey

Saltanat T. Urazayeva[1], Saulesh S. Kurmangaliyeva[2], Asset A. Kaliyev[3], Kymbat Sh. Tussupkaliyeva[1], Arman Issimov[4]*, Aisha B. Urazayeva[1], Zhuldyz K. Tashimova[1], Nadiar M. Mussin[3], Toleukhan Begalin[1], Aimeken A. Amanshiyeva[1], Gulaiym Zh. Nurmaganbetova[1], Shara M. Nurmukhamedova[1], Saule Balmagambetova[5]*

1 Department of Epidemiology, West Kazakhstan Marat Ospanov Medical University, Aktobe, Kazakhstan, 2 Department of Microbiology, Virology and Immunology, West Kazakhstan Marat Ospanov Medical University, Aktobe, Kazakhstan, 3 Department of General Surgery, West Kazakhstan Marat Ospanov Medical University, Aktobe, Kazakhstan, 4 Department of Biology, K.Zhubanov Aktobe Regional University, Aktobe, Kazakhstan, 5 Department of Oncology, West Kazakhstan Marat Ospanov Medical University, Aktobe, Kazakhstan

* issimovarman@gmail.com (AI); saule.balmagambetova@zkmu.kz (SB)

## Abstract

The scale of emergency caused by COVID-19, the ease of survey, and the crowdsourcing deployment guaranteed by the latest technology have allowed unprecedented access to data describing behavioral changes induced by the pandemic. The study aimed to present the survey results identifying attitudes toward vaccination against COVID-19 among the population of West Kazakhstan, the level of confidence in the national QazVac vaccine, and the role of different sources of information on COVID-19 in decision-making concerning vaccination. A computer-assisted survey was conducted using WhatsApp messenger. Overall, 2,009 participants responded, with a response rate of 92%. Most (83.1%) were immunized against COVID-19; among them, 20.1% obeyed the request of their employers that had been practiced within non-pharmaceutical interventions to contain the disease. The youngest respondents, individuals with a college education, students, and employed people, as well as those with chronic diseases, showed positive attitudes toward vaccination (all p<0.05). About two-thirds of respondents (69.2%) expressed trust in all types of vaccines against COVID-19. Of those who refused vaccination (16.9%), about one-third feared vaccination consequences, and more than a third (38.2%) reported anti-vaccine sentiments. The decisive factors in accepting vaccination were trust in official sources of information (reports of medical experts, etc.) and, mainly, subjectively interpreted sufficiency of information about the disease, which had increased the odds of being vaccinated by 63.9% (OR 1.71, 95% CI [1.3;2.26], p<0.05). Confidence in the domestic QazVac vaccine was expressed by 37.7% of respondents. History and severity of COVID-19 disease did not play a role in positive perceptions of vaccination, while illness after vaccination substantially affected vaccination approval (p 0.021). No significant differences have been observed regarding the overall

**Data Availability Statement:** All relevant data are within the manuscript and its Supporting Information files.

**Funding:** This research was funded by the Science Committee of the Ministry of Science and Higher Education of the Republic of Kazakhstan (Grant No. AP14870878). The funders had no role in the design of the study; in the collection, analyses, or interpretation of data; in the writing of the manuscript, or in the decision to publish the results.

**Competing interests:** The authors have declared that no competing interests exist

performance across five vaccines (QazVac, Sputnik V, CoronaVac, Hayat-Vax, and BioN-Tech/Pfizer) available for Kazakhstan's population (p 0.27).

## Introduction

The World Health Organization (WHO) first declared that SARS-CoV-2 infection had become a pandemic on March 11, 2020, when 118,319 infections and 4,292 deaths were reported in 113 countries [1]. The pandemic constituted a severe test for societal life worldwide and significantly affected public health. During this period, non-pharmaceutical interventions (NPIs) have been one of the key weapons against the SARS-CoV-2 virus, affecting virtually any societal process. It has been proven that school closing, followed by workplace closing, business and venue closing, and public event bans were the most effective NPIs in controlling the spread of COVID-19 [2, 3]. Government effectiveness and the rule of law increased the efficiency of NPIs, while regulatory quality was the most crucial dimension of their efficacy [4].

Vaccination policy was another critical tool for national governments to reduce the spread of the virus, and the World Health Organization (WHO) strongly supported the use of the approved COVID-19 vaccines even though no vaccine had been 100% protective [5]. In 2019, the WHO announced vaccine hesitancy as one of the ten threats to global health. Experts have identified several common reasons for vaccine hesitancy, including the belief that a person is not at risk for the disease, limited access to vaccines, and lack of trust in their safety and efficacy [6]. A study by Global Market Research and Public Opinion Specialist (IPSOS), which surveyed 18,526 respondents from 15 countries worldwide, found that 73% of respondents were willing to be vaccinated against COVID-19 [7]. According to data from the London-based YouGov Data Center, by 2021, the UK had the highest proportion of respondents willing to receive the COVID-19 vaccine, at 71.3%, while France had the lowest, at 29.8% [8].

Around the same period, the Central Asia Regional Economic Cooperation (CAREC) studied attitudes towards COVID-19 vaccination in seven countries, including Kazakhstan. While most respondents expressed positive opinions about the effectiveness of vaccinations, the lowest proportion (29.2%) was found among respondents from Kazakhstan [9]. Generally, cross-country differences concerning vaccination willingness can be explained by the fact that decisions about vaccination are based on more than mere knowledge of risks, costs, and benefits. Individual decision-making about vaccinating involves many other factors, including emotion, culture, religion, and socio-political context [10]. In the meantime, Kazakhstan was one of the countries where the inactivated vaccine QazCovid-in (QazVac) had been developed. During a 6-month follow-up period, the QazVac vaccine showed 82% protective efficacy (95% CI 71.1;88.5) in volunteers [11, 12].

Cases of the disease in Kazakhstan have been registered since March 2020; in the Aktobe region, the first imported case from abroad was noted on 22.03.2020. Since the beginning of the pandemic, 1,408,124 confirmed cases of COVID-19 infection have been registered in Kazakhstan, 13,846 fatal cases, and 11,258 patients have been receiving treatment (as of March 14, 2023). There is no proof of fabrication for Central Asian republics reported COVID-19 cases [4]. Kazakhstan, like other countries, also introduced non-pharmaceutical interventions. Researchers who explored the strategies applied in the Central Asian region to contain COVID-19 concluded that, except for Turkmenistan, the rest of the region states adopted a very similar approach and that deaths, more than cases, pushed governments to impose restrictions [13].

Vaccination started in February 2021, and mandatory vaccination was required for health providers, teachers of secondary schools, and police officers; then, as access to vaccines expanded, it became compulsory for students and employees of higher and secondary educational institutions. Several types of vaccines were available to the public: Kazakhstan's QazVac, Russian Sputnik V and Sputnik Lite, Chinese Hayat-Vax (Sinopharm) and CoronaVac, and later Pfizer's Comirnaty vaccine (USA) for vaccination of adolescents, pregnant women and patients of maternity wards [14]. It is commonly accepted that several types of vaccines allow for more rapid collective immunity. Vaccination against COVID-19 was free for all population categories; a second vaccination was recommended 9–12 months after the first round.

Overall, the emergency scale, ease of survey, and crowdsourcing deployment guaranteed by the latest technology have allowed unprecedented access to data describing behavioral changes induced by the pandemic [3]. During the pandemic and after its official termination announced in 2022, much research has been published on various aspects of COVID-19. At the same time, few studies in Kazakhstan and none across the country's western region have examined behavioral patterns during the pandemic, attitudes toward vaccination, and the reasons for the population's approval or refusal of vaccination. Our research aims to fill this gap.

Thus, the study aimed to present the survey results to identify attitudes toward vaccination against COVID-19 among the population living in Aktobe, a sizeable western city. We also explored the confidence level in the national QazVac vaccine and the role of different sources of information on COVID-19 in individual decision-making about vaccination.

## Materials and methods

### Ethics statement

This study posed no risk to the participating individuals. Survey participation did not imply restrictions on receiving any clinical care determined by specialists. The study was conducted according to the principles of the Declaration of Helsinki (2013), and patient rights were observed. The purpose and methods of the current study were explained to all participants in the preamble to the questionnaire and in the informed consent form, which was obtained separately from each respondent through WhatsApp messenger. All respondents who filled out the questionnaire checked the consent box. Participation in the survey was voluntary and confidential, and these provisions were stated in the informed consent form and the preamble to the questionnaire. The respondents were informed that their data would be placed anonymously in the research database without mentioning personal details, i.e., using coded numbers, and their confidential information would not be disclosed when publishing the results of this survey. All participants gave their consent to publishing the results of the study. The research protocol was approved by the Bioethics Committee of the West Kazakhstan Marat Ospanov Medical University (No. 7–07, ref. 7, dated 28.09.2022.)

### Study design and participants

A cross-sectional survey was conducted from October 28 to November 30, 2022, among residents of Aktobe, the largest city in West Kazakhstan (549,381 population as of 2022). The survey was computer-assisted through the Google platform. Adult persons who decided to participate in the study were enrolled using the convenience sampling technique. The assigned researchers distributed the questionnaire among Aktobe residents through the following personal WhatsApp groups (chats): "Friends," "Relatives," "Colleagues," "Students," and "Acquainted persons." The informed consent form was sent separately in each chat and obtained from subjects together with the filled questionnaires. The inclusion criteria involved Aktobe residents aged 18–75 without mental disorders from different social groups. As

designated researchers from the project staff distributed the questionnaire among their surroundings, individuals with mental disorders had no opportunity to participate. Other exclusion criteria were persons under 18 or older than 75 and those living out of Aktobe. No limitations were implied by gender, ethnicity, social and employment status, education level, presence of chronic somatic diseases, or vaccination status. The authors did not use incentives or advertising to recruit participants. WhatsApp Messenger has been chosen because it is the most frequently used network by the population in Kazakhstan.

## Data collection

We designed and used a data collection questionnaire with various closed-ended questions, implying dichotomous or multiple-choice responses and including those in a Likert scale. After 15 participants were fully informed about the study and signed the informed consent form, the pilot launching was conducted; these participants were then excluded from the study. Five questionnaire iterations were developed and tested, involving 15 participants each time, until the acceptable validation level, Cronbach alpha 0.7, was reached. The questionnaire was prepared so the participants could easily and independently complete it. The design of the questionnaire provided the respondents with anonymity and confidentiality protection. No identifying (personal) information was requested or recorded, even the exact birth date, only age groups. The experts on the questionnaire content were epidemiologists involved in arranging and promoting vaccination of the population against COVID-19 and university scientists with experience in questionnaire development methodology. The questionnaire was developed in two languages, Russian and Kazakh, for the convenience of the respondents' choice of language.

The questionnaire consisted of three domains (see S1 Table):

A. The demographic domain included six questions about age, gender, education, type of activity, monthly income per capita, and somatic chronic diseases (presence/absence.) The "Education" item implied four levels: incomplete secondary, secondary (school), secondary professional (presence of a specialty that does not require a university diploma—electrician, plumber, highly qualified worker, etc.), and higher education (university diploma.) The "Type of activity" item included four categories: nonworking (unemployed, housewives), student, employed, and retired. The "Income" item included three extents of monthly income per capita: high (500 thousand tenges (national currency) and above, which is approximately equal to 1,000 US dollars); middle (from 100 to 500 thousand tenges, i.e., in the range of 200–1,000 US dollars), and low income (100 thousand tenges and below, i.e., 200 US dollars per month or less.) The item "Presence/absence of somatic chronic diseases" included five response options—the absence of disease, arterial hypertension (AG), coronary heart disease (CHD), diabetes mellitus (DM), and other diseases that should be inscribed.

B. "Attitude/Trust in Vaccination Against COVID-19 and Information Sources" domain contained eight questions:

- What type of vaccine against COVID-19 did you receive last time?

- What was the reason for getting vaccinated?

- Why have you refused vaccination if so?

- Which sources of information do you trust?

- In your opinion, has enough information about COVID-19 vaccination been provided in official sources?

- Which COVID-19 vaccine do you trust?

- What is your attitude to vaccination against COVID-19? (Likert scale responses)

- What is your level of confidence in Kazakhstan's QazVac vaccine? (Likert scale responses)

C. "Questions Concerning the Experience with COVID-19" domain included six questions:

- Have you been ill with COVID-19?

- By what method of testing have you been diagnosed with COVID-19 disease?

- In what kind of condition did you suffer from the COVID-19 disease?

- Have you been ill with COVID-19 disease after vaccination?

- How long has a doctor observed you after suffering COVID-19?

- Which of the following symptoms did you have after the disease?

## Statistical analysis

The Kolmogorov-Smirnov test was applied to determine the data distribution, skewness, and kurtosis coefficients in the present study, where data were not normally distributed. Sociodemographic characteristics were defined using descriptive statistics (frequency, percentage). Demographic data and responses to the questionnaire were treated as categorical variables (the reasons for getting vaccinated, refusing vaccination, etc.) The Pearson chi-squared criterion was applied to evaluate the relationships between the independent variables (demographic data, level of trust in different information sources, etc.) and attitudes toward vaccination, thus identifying intergroup differences. In evaluating the level of confidence in the national QazVac vaccine across various groups and investigating whether there was a statistically significant difference between this variable of interest and the sub-categories of the other variables (sociodemographic, etc.), the Kruskal-Wallis rank sum test was applied. To assess the influence of independent factors (obtaining information about COVID-19 vaccination; the amount of information regarding vaccines) on the binary variable of response (being vaccinated; trust in the Qazvac vaccine), logistic regression analyses (LRA) were performed. Getting a vaccine and trusting in the Qazvac vaccine were treated as a "positive" effect, and the opposite was treated as a "negative" effect. Variables were included if $p < 0.1$ in the univariate analysis (chi-squared test.)

The sample size calculation was performed, considering the expected logistic regression analysis. As stated in the Guidelines on sample size calculation for logistic regression [15], a minimum sample of 500 is required to conduct a logistic regression properly. Besides, considering the number of independent variables in the final logistic regression model (i), the minimum sample size ($N$) can be estimated as $N = 100 + 50i$.

Two-sided levels at 0.05 were assumed to be statistically significant. For statistical processing, software packages SPSS (IBM, Armonk, NY, USA, v.25) and Statistica (Stat-Soft, Inc., Tulsa, OK, USA, v. 10) were used.

## Results

Overall, questionnaires were sent out to 2,184 individuals, and 2,009 participants responded, with a response rate of 92%. Out of 2,009 interviewees, the majority were women, and the most common age range was 18–29. By occupational status, most were students or employed. Among the participants, almost two-thirds had a college education and a middle-income level (Table 1).

**Table 1. Demographic characteristics, attitude/trust in vaccination against COVID-19, and information sources used by respondents.**

| | Parameters (questions) | N 2,009 (%) | Vaccinated 1,669 | Not vaccinated 340 | Pearson's χ2 | P |
|---|---|---|---|---|---|---|
| 1. | Gender: | | | | χ0.005 | 0.98 |
| | Men | 643 (32.0%) | 534 (32%) | 109 (32.1%) | | |
| | Women | 1,366 (68.0%) | 1,135 (68%) | 231 (67.9%) | | |
| 2. | Age: | | | | 1.63 | 0.65 |
| | 18–29 | 972 (48.4%) | 807 (48.4%) | 165 (48.5%) | | |
| | 30–39 | 428 (21.3%) | 349 (20.9%) | 79 (23.2%) | | |
| | 40–59 | 403 (20.1%) | 342 (20.5%) | 61 (17.9%) | | |
| | 60–75 | 206 (10.3%) | 171 (10.2%) | 35 (10.3%) | | |
| 3. | Occupation: | | | | 65.3 | 0.001 |
| | Nonworking | 175 (8.7%) | 125 (7.5%) | 50 (14.7%) | | |
| | Student | 936 (46.6%) | 801 (48%) | 135 (39.7%) | | |
| | Employed | 825 (41.1%) | 703 (42.1%) | 122 (35.9%) | | |
| | Retired | 73 (3.6%) | 40 (2.4%) | 33 (9.7%) | | |
| 4. | Education: | | | | 34.7 | 0.001 |
| | Incomplete secondary | 42 (2.1%) | 29 (1.7%) | 13 (3.8%) | | |
| | Secondary (school) | 335 (16.7%) | 253 (15.2%) | 82 (24.1%) | | |
| | Secondary special | 361 (18.0%) | 286 (17.1%) | 75 (22.1%) | | |
| | College (university) | 1,271 (63.3%) | 1,101 (66%) | 170 (50%) | | |
| 5. | Monthly income per capita: | | | | 6.13 | 0.047 |
| | High | 299 (14.9%) | 244 (14.6%) | 55 (16.2%) | | |
| | Middle | 1,311 (65.3%) | 1,108 (66.4%) | 203 (59.7%) | | |
| | Low | 399 (19.9%) | 317 (19%) | 82 (24.1%) | | |
| 6. | 1. Presence of chronic diseases (AG, CHD, DM, other)[1] | | | | 3.47 | 0.063 |
| | | 268 (13.3%): | 212 (12.7%) | 56 (16.5%) | | |
| | 2. No chronic diseases | 1,741 (86.7%) | 1,457 (87.3%) | 284 (84.5%) | | |
| 7. | What type of vaccine against COVID-19 did you receive last time?[2] | | | | | |
| | 1. Sputnik V | 763 (45.7%) | | | | |
| | 2. QazVac | 305 (18.3%) | | | | |
| | 3. Hayat-Vax (Sinopharm) | 96 (5.8%) | | | | |
| | 4. CoronaVac (Sinovac) | 64 (3.8%) | | | | |
| | 5. BioNTech/Pfizer | 195 (11.7%) | | | | |
| | 6. I do not remember the name of the vaccine | 246 (14.7%) | | | | |
| | 7. I did not take any[3] | 340 (16.9%) | | | | |
| 8. | The reason for getting vaccinated[2] | | | | | |
| | 1. At my own will | 1,089 (65.3%) | | | | |
| | 2. At employer's request | 337 (20.1%) | | | | |
| | 3. Based on the recommendation of medical workers | 243 (14.5%) | | | | |
| | 4. Refused vaccination[3] | 340 (16.9%) | | | | |
| 9. | Why did you refuse vaccination if yes?[3] | | | | | |
| | 1. I'm afraid of complications after vaccination | 93 (27.4% out of 340) | | | | |
| | 3. I am against all vaccines | 65 (19.1%) | | | | |
| | 3. I do not believe in the effectiveness of COVID-19 vaccines | 65 (19.1%) | | | | |
| | 4. I have medical contraindications | 71 (20.9%) | | | | |
| | 5. My religion does not allow it | 46 (13.5%) | | | | |
| | 6. I received a vaccine | 1,669 (83.1%) | | | | |

(*Continued*)

**Table 1.** (Continued)

| | Parameters (questions) | *N* 2,009 (%) | Vaccinated 1,669 | Not vaccinated 340 | Pearson's χ2 | *P* |
|---|---|---|---|---|---|---|
| 10. | Which sources of information do you trust? | | | | | |
| | 1. Official sources (media, medical professionals) | 1,125 (56.0%) | 1,111 (66.5%) | 93 (27.3%) | 44.4 | 0.001 |
| | 2. Unofficial sources (family members, colleagues) | 418 (20.8%) | 267 (16%) | 70 (20.6%) | | |
| | 3. Others (speeches of famous personalities, social networks–WhatsApp, YouTube) | 466 (23.2%) | 291 (17.4%) | 177 (52.1%) | | |
| 11. | In your opinion, has enough information about COVID-19 vaccination been provided in official sources? | | | | | |
| | 1. Few | 408 (20.3%) | 212 (12.7%) | 196 (57.7%) | 92.3 | <0.001 |
| | 2. Sufficient | 1,357 (67.5%) | 1,256 (75.2%) | 101 (29.7%) | | |
| | 3. Many | 125 (6.2%) | 111 (6.7%) | 14 (4.1%) | | |
| | 4. A vast amount of information | 119 (5.9%) | 90 (5.4%) | 29 (8.5%) | | |
| 12. | Which COVID-19 vaccine do you trust?[4] | | | | | |
| | 1. I do not trust all vaccines | 619 (30.8%) | | | | |
| | 2. National vaccine (QazVac) | 251 (758) | | | | |
| | 3. Neighboring countries (Sputnik V) | 305 (541) | | | | |
| | 4. Far-abroad countries (Hayat-Vax, CoronaVac, BioNTech/Pfizer) | 250 (454) | | | | |
| | 5. Combined responses regarding vaccine preferences | 584 | | | | |
| | 6. I trust all vaccines | 1,390 (69.2%) | | | | |
| 13. | Your attitude toward vaccination against COVID-19 | | | | | |
| | 1. Positive | 1,298 (64.6%) | 1,251 (74.9%) | 47 (13.8%) | 246.8 | <0.001 |
| | 2. Indifferent | 92 (4.6%) | 85 (5.1%) | 7 (2.1%) | | |
| | 3. Negative | 619 (30.8%) | 333 (20%) | 286 (84.1%) | | |
| 14. | What is your level of confidence in Kazakhstan's QazVac vaccine? | | | | | |
| | 1. I do not trust | 1,251 (62.3%) | 979 (58.7%) | 272 (80.0%) | 54.8 | 0.001 |
| | 2. I trust partially | 257 (12.8%) | 232 (13.9%) | 25 (7.4%) | | |
| | 3. I trust | 427 (21.3%) | 387 (23.2%) | 40 (11.8%) | | |
| | 4. I trust completely | 74 (3.7%) | 71 (4.3%) | 3 (0.9%) | | |
| 15. | Have you been ill with COVID-19?[5] | | | | | |
| | 1. No | 1,233 (61.4%) | 1,028 (61.6%) | 205 (60.3%) | 2.51 | 0.285 |
| | 2. Not sure (I was more likely ill) | 374 (18.6%) | 301 (18.0%) | 73 (21.5%) | | |
| | 3. Yes | 402 (20.0%) | 340 (20.4%) | 62 (18.2%) | | |

Note

[1]AG—Arterial hypertension; CHD—Coronary heart disease; DM—Diabetes mellitus.

[2]In the case of getting a vaccine, a percentage is indicated out of 1,669, i.e., out of those who received a vaccine.

[3]In refusing a vaccine, a percentage is indicated out of 340, i.e., out of those who refused a vaccine.

[4]Multiple answers to a specific question were received on this item. Chi-sqr. calculations through groups of vaccines were not possible.

[5]This item means before vaccination

## Differences between vaccinated and unvaccinated respondents

We analyzed the sample by vaccination status. Of 2,009 participants, 1,669 (83.1%) received the vaccine, and 340 (16.9%) refused vaccination. Table 1 displays the descriptive and comparative statistics regarding the differences between vaccinated and unvaccinated interviewees.

Gender, age and the presence of chronic diseases did not influence the decision to be vaccinated (p<0.05, Table 1). Most interviewees suffering from chronic diseases (268, or 13.3% out of 2,009) provided combined answers on item 6. Among these 268 individuals, 73 indicated the presence of AG+CHD (27.2%), 50 respondents marked AG+DM (18.7%), 34 noted CHD +DM (12.7%), 31 (11.5%) pointed all the listed diseases, and 80 (29.9%) indicated only one somatic disease, respectively. Almost four out of five suffering from chronic diseases (212 out of 268, 79.1%) decided to get a vaccine. The proportion of participants with no somatic diseases was approximately similar among those vaccinated and those who refused. Notably, a previous (before vaccination and before the survey) history of COVID-19 disease also did not affect the decision to get vaccinated. Individuals who had not been ill with COVID-19 and those who had been ill or thought themselves sick were approximately equally divided among immunized and unimmunized respondents (p 0.285, Table 1).

The factors that significantly impacted vaccination decisions appeared to be the level of education, occupational status, and interest in getting information on COVID-19 issues through available sources. Of those who were vaccinated, 66% had a higher education. The majority of vaccinated were students (48%) and employed people (42.1%). Nonworking individuals and respondents of retirement age were more likely to refuse immunization (14.7% vs. 7.5% and 9.7% vs. 2.4% of those vaccinated, respectively). Regarding the income level, the highest proportion of those who preferred to take preventive measures against COVID-19 should have been expected among individuals with high incomes. However, respondents with a middle-income level (200–1,000 USD per capita) provided the majority of immunizations (66.4%, p 0.047). In the meantime, 20.1% of respondents indicated they had been subject to a definite degree of pressure from their employers in deciding to be vaccinated. About two-thirds (65.3%) of participants decided independently.

## The role of information sources in the decision-making regarding vaccination by respondents

Significant differences between vaccinated and unvaccinated participants were observed concerning how to get information on the disease (p 0.001). Of all vaccinated respondents (1,669), two-thirds (66.5%) trusted official sources of information about the disease, while 72.7% of those who refused (N 340) trusted alternative sources (item 10, Table 1). The majority of vaccinated interviewees (75.2%) found the available information "sufficient," while 57.7% of those who refused (340) considered the information "scarce" (p<0.001).

We utilized a simple binary logistic regression model to determine the predictors of favorable vaccination decisions. A total of 8 variables with p<0.1 were present in the univariate analysis (Table 1). The multivariate LRA of these variables showed that, after adjusting, only one variable (the amount of relevant information) contributed significantly to the participants' willingness to be vaccinated (p-log<0.05). Our analysis presented a chi-square statistic ($\chi^2(1) = 18.6$, p<0.001), refuting the null hypothesis and signifying that the "sufficient" amount of available information predicts the decision to get a vaccine (S2 Table). The model's R2 value, standing at 0.15, suggests that variations in the amount of information account for approximately 15% of the variance in the final decision, highlighting the influence of "sufficient" available information about the disease in official sources on the eventual decision of getting a vaccine in the respondents. The odds ratio (OR) calculated at 1.71, with a 95% CI of [1.3;2.26], p < 0.05, elucidates that a "sufficient" amount of relevant information increases the odds of being vaccinated by 63.9%. When the respondents assess the amount of information as "many" or "too many (excessive)," the odds of deciding on vaccination turn insignificant: OR 1.36, 95% CI of [0.81;2.28], p 0.24, and OR 0.9, 95% CI of [0.56;1.45], p 0.68, respectively.

## Attitudes toward vaccination among the West Kazakhstan population

In item 13 (Table 1), we presented 1,298 interviewees who positively treated vaccination, while 711 individuals treated indifferently (92) or negatively (619). The most significant contribution to the positive attitudes toward vaccination was made by persons vaccinated (74.9%), whereas 86.2% of those who refused (340) treated vaccination indifferently or negatively (chi-sqr. 248.6, p<0.001).

We also analyzed the relationship between the acceptance of a vaccine against COVID-19 and other parameters irrespective of the vaccination status of interviewees (S3 Table). Gender, income level, and history of COVID-19 (whether respondents were ill or not) did not influence the vaccination approval (p<0.05, S3 Table). Of those who welcomed vaccination, the most significant contributions were made by persons aged 18–29 (61.2%, p 0.001), individuals with a college education (64.8%, p 0.001), and students (48.4%, p 0.02). In comparison, non-working individuals were primarily negative or indifferent (10.7% vs. 7.6% of those who treated positively). Those who suffered from chronic diseases were undoubtedly positive toward vaccination (192 vs. 76, p 0.01), though this sample showed other statistics in Table 1 (N 212 vaccinated vs. 56 unvaccinated). Individuals who welcomed vaccination trusted official sources of information about COVID-19 (66.4%), while 31.9% of those with a negative attitude toward vaccination demonstrated trust in various YouTube podcasts (p 0.022). Of 1,669 immunized individuals, 66.5% trusted official information about the disease, while among those who refused vaccination (N 340), more than 52.1% relied on alternative sources (Table 1). About three-quarters of those who positively treated vaccination found the available official information "sufficient" (74.3%, p 0.001). They also welcomed the national QazVac vaccine (54.7% vs. 6.8% of those who did not trust it, p 0.001).

Thus, studying the general attitude towards SARS-CoV-2 immunization among the population, regardless of vaccination presence/absence in interviewees, confirmed the general trends and indices identified in the analysis across vaccination status (able 1) despite insignificant discrepancies.

In a separate analysis of the sample of those who refused vaccination (N 340), we found no differences in gender, age, and income (p>0.05). Still, nonworking people and those with low incomes prevailed compared to the vaccinated. About one-third (27.5%) feared vaccination consequences, and 38.2% expressed anti-vaccine sentiment (19.1% lack confidence in vaccines' effectiveness, and 19.1% lack trust in vaccines in general). In item 12 (Table 1), the proportion of respondents who stated their lack of confidence in vaccines was 30.8%. Accordingly, 80% of those who refused vaccination did not trust the national QazVac vaccine (item 14).

## The course of the disease in those who fell ill with COVID-19

In Table 1, item 15, we reported the total number of those who did not face COVID-19 before vaccination (N 1,233 out of 2,009, or 61.4%) and those who doubted whether they fell ill with COVID-19 (likely fell ill, 374, or 18.6%). These individuals did not consult a doctor to establish a diagnosis. Still, they experienced illness during the pandemic, which allows them to be conditionally classified as having recovered from COVID-19. The number of those who suffered from the disease with an established diagnosis was 402, or 20%. The descriptive statistics regarding the details of COVID-19 in respondents are displayed in Table 2.

Figuring out precise statistics on the incidence of COVID-19 among the respondents appeared to be difficult, as many of them could not see a doctor during the pandemic. Approximately one-fifth of them considered themselves to be those who fell ill (N 374) without an established diagnosis and subsequent follow-up by a doctor. The same applies to those who thought themselves ill after vaccination (N 287). PCR tests, chest X-rays, and CT scans were the most prevalent methods of COVID-19 diagnosis (Table 2). Overall, 776 individuals (38.6%

**Table 2. Descriptive statistics of the disease course in those who suffered from COVID-19.**

| Item | Parameters (questions) | | | |
|---|---|---|---|---|
| | Answers details | Abs (%) | Combinations | Abs (%) |
| 16. | By what method of testing have you been diagnosed with COVID-19 disease?* | | | |
| | 1. PCR test | 266 | 1. CT scan+Doctor | 68 |
| | 2. Chest X-ray / CT scan | 232 | 2. PCR+CT scan | 62 |
| | 3. ELISA for antibodies | 142 | 3. PCR+Doctor | 58 |
| | 4. During the doctor's examination | 140 | 4. PCR+ELISA | 37 |
| | | | 5. ELISA+Doctor | 28 |
| | | | 6. ELISA+CT scan | 26 |
| 17. | In what kind of condition did you suffer from the COVID-19 disease? | | | |
| | 1. Was not ill | 1,233 (61.4% out of 2,009) | | |
| | 2. Mild form | 371 (47.8% out of 776—who were ill or thought themselves ill) | | |
| | 3. Moderate form | 341 (43.9% out of 776) | | |
| | 4. Severe form | 64 (8.3% out of 776) | | |
| | Have you been ill with COVID-19 disease after vaccination? | | | |
| | 1. I have not received a vaccine | 340 (16.9% out of 2,009) | | |
| | 2. I was not ill | 1,210 (72.5% out of 1,669 vaccinated) | | |
| | 3. Not sure (more likely ill) | 287** (17.2% out of 1,669) | | |
| | 4. I was ill | 172 (10.3% out of 1,669) | | |
| 19. | How long have you been observed by a doctor after suffering COVID-19? | | | |
| | 1. Was not observed by a doctor (as the disease was not being diagnosed) | 1,607 (1,233+374) | | |
| | 2. 3 months | 150 (37.3% out of 402) | | |
| | 3. 6 months | 171 (42.5% out of 402) | | |
| | 4. 12 months (1 year) or more | 81 (20.2% out of 402) | | |
| 20. | Which of the following symptoms did you have after the disease? | | | |
| | 1. Violation of the sense of smell or taste | 394 | 1. Violation of the sense of smell or taste+Fatigue | 332 |
| | 2. Decreased memory and mental performance | 319 | 2. Violation of the sense of smell+Decreased memory | 203 |
| | 3. Sleep disorder | 311 | 3. Decreased memory +Cough | 192 |
| | 4. Cough | 293 | 4. Violation of the sense of smell or taste+Hair fall | 191 |
| | 5. Shortness of breath | 276 | 5. Decreased memory+Sleep disorder | 185 |
| | 6. Depression | 213 | 6. Violation of the sense of smell or taste+Joint pain | 169 |
| | 7. Hair fall | 210 | 7. Decreased memory+Joint pain | 159 |
| | 8. Joint pain | 176 | 8. Decreased memory+Depression | 139 |
| | 9. Allergy | 131 | 9. Decreased memory+Shortness of breath | 136 |
| | 10. Rash | 100 | 10. Shortness of breath+Violation | 110 |
| | 11. Fatigue | 92 | 11. of the sense of smell or taste | |
| | 12. Others (inscribe) | - | | |

Note

* The total number of those who have been ill with an established diagnosis was 402.

**These individuals could not clarify whether they fell ill with COVID-19 after vaccination. Still, they experienced a set of symptoms, which allows them to be conditionally classified as having recovered from the disease.

out of 2,009) reported a history of COVID-19 infection with severities of different degrees (402 had an established diagnosis and were followed up). Among these 776 convalescents, more than half (52.2%) experienced moderate and severe forms of the disease with various manifestations of likely post-COVID syndrome.

Violation of the sense of smell and taste, frequently combined with fatigue, as well as decreased memory and mental performance, and sleep disorders were the most prevalent symptoms in this group of interviewees. The presence of multiple combined answers to a specific question does not allow for performing comparative statistics within this questionnaire section.

## Attitudes toward the national QazVac vaccine

Overall, 69.2% of respondents trusted all vaccines (item 12, Table 1), while vaccination against COVID-19 was treated positively by 64.6% (item 13). The negative attitude to all vaccines and vaccination in general was expressed by 30.8% of interviewees. Attitudes toward all vaccines, positive or negative, were not affected by the experience of COVID-19 illness of varying severity (p 0.65, S4 Table).

Many respondents chose multiple answers regarding trust in the definite vaccine type (item 12, Table 1). Of 1,390 respondents who trusted all vaccines, 806 provided single clear answers regarding the choice of specific vaccines, while 584 interviewees preferred combined answers. Of them, 334 (57.2%) highlighted a combination of all three groups of vaccines, 130 (22.3%) marked a combination of QazVac and vaccines from neighboring countries, 77 (13.2%) preferred a combination of vaccines from neighboring and far-abroad countries, and 43 (7.3%) pointed out QazVac + far-abroad vaccines. In total, 758 (37.7%) respondents marked QazVac separately and in different combinations. They expressed various extents of their favorable attitudes towards QazVac (item 14), from partial trust (12.8%) to complete acceptance (3.7%), albeit only 305 were immunized with this vaccine (item 7, Table 1). There were no significant differences between supporters and those who were distrustful of the QazVac vaccine across their education level (p 0.23), income (p 0.7), occupational status (p 0.42), presence/absence of chronic diseases (p 0.071), and personal history of COVID-19 disease (p 0.83, S5 Table). It is noteworthy that significant differences emerged by gender. Most men welcomed the domestic vaccine (52.7% of supporters vs. 19.5% of men treated negatively and 47.3% vs. 80.5% of women, respectively, p 0.001, chi-sqr. 26.9). In terms of age, 36.1% of those aged 30–39 trusted in QazVac vs. 10.3% of their opponents. At the same time, a substantial proportion of the youngest population (18–29 yrs.) treated Kazakhstan's vaccine mostly negatively (35.1% of supporters vs. 56.4% of opponents, p 0.004). The age differences were also revealed between the youngest and oldest respondents within the groups of supporters and opponents of the national vaccine, $H$ (3, $N$ 2,009, R = 1,094.3, p 0.01).

Supporters and those skeptical toward the domestic vaccine differed significantly in how they obtained information and the amount of information necessary to express judgments. Almost seven out of ten who approved introducing the national vaccine (69.9%) trusted the official sources of information vs. 47.6% of those who were pessimistic about the domestic product (p 0.018). Three-quarters of supporters (75.1%) considered the information regarding COVID-19 and QazVac "sufficient" for trust (chi-sqr. 46.1, p 0.001). We designed a binary logistic regression model to reveal the possible predictors of trust in the national vaccine. A total of 5 variables with p<0.1 were present in the univariate analysis (S5 Table). After adjusting for the other variables, the multivariate analyses resulted in the presence of one variable (the amount of relevant information) that contributed significantly to the participants' trust in the national vaccine (p-log<0.05). A chi-square statistic $\chi^2(1)$ of 48.4 p<0.001 signifies that "sufficient" information in official sources predicts trust in the domestic vaccine. The model's R2 value of 0.32 suggests that variations in the amount of information account for approximately 32% of the variance in the final result, highlighting the influence of available information about the disease and a vaccine of interest in official sources on the judgment about the

vaccine in the respondents. The OR 2.35, with a 95% CI of [1.82;3.02], $p < 0.05$, evidence that a "sufficient" amount of relevant information increases the odds of trust in the QazVac vaccine by 70.1%. Yet, the likelihood of confidence in the vaccine decreases with further increments up to the level of "many information" (OR 2.24, 95% CI [1.47;3.42], p 0.001) and turns insignificant at the level of "superfluous information" (OR 1.77, 95% CI [1.14;2.75], p 0.11).

## COVID-19 cases after vaccination and the overall effectiveness of vaccines in respondents

Previously, we analyzed attitudes toward vaccination among the general sample (N 2,009). As for 1,669 vaccinated respondents, a total of 1,188 (71.2%) had a positive perception of vaccination, and 481 (28.8%) expressed indifferent or negative attitudes (S7 Table). Significant differences were identified when analyzing from the perspective of those who experienced COVID-19 after vaccination (p 0.021, S7 Table). Most of those 1,210 vaccinated who did not get sick after vaccination approved immunization (74.4% versus 67.9% of those who were negative or indifferent). However, individuals who suffered from the disease after vaccination expressed predominantly negative sentiments. Among those 172 who experienced moderate or severe forms of COVID-19 after vaccination, 12.5% were unfriendly toward vaccines, vs. 9.4% who remained their supporters. Among those 287 who did not have a diagnosis but considered themselves to have recovered from the disease, the same proportion was observed: 19.8% skeptics vs. 16.2% supporters of vaccination (S7 Table).

We performed an analysis of the relationship between the type of vaccine received and the incidence of COVID-19 after vaccination, using data from item 7 ("What type of vaccine against COVID-19 did you receive?") and item 18 ("Have you been ill with COVID-19 disease after vaccination?"). Table 3 presents the results: 72.5% (1,210) of all vaccinated individuals did not get COVID-19 after vaccination, while 459 (27.5%) reported a history of illness after vaccination. Of them, only 172 (10.3% out of 1,669 vaccinated) clearly indicated the COVID-19 disease after vaccination (the descriptive statistics on item 18 are also presented in Table 2).

Among people vaccinated with specific vaccines, we obtained the following rates of those who avoided COVID-19: Sputnik V, with 572 individuals out of 763 (75%) not falling ill with

**Table 3. Analysis of the relationship between the type of vaccine received and the presence of the disease after vaccination.**

| Parameters/ Items | N 1,669 (vaccinated) | Those who were ill after vaccination, N 459* | Those who were not ill after vaccination, N 1,210 | Pearson's χ2 | P |
|---|---|---|---|---|---|
| What type of vaccine against COVID-19 did you receive last time? | 1. Sputnik V | | | χ6.43 | 0.27 |
| | 763 (45.7) | 191 (41.6%) | 572 (47.3%) | | |
| | 2. QazVac | | | | |
| | 305 (18.3) | 94 (20.5%) | 211 (17.4%) | | |
| | 3. Hayat-Vax | | | | |
| | 96 (5.8%) | 27 (5.9%) | 69 (5.7%) | | |
| | 4. CoronaVac | | | | |
| | 64 (3.8%) | 23 (5.0%) | 41 (3.4%) | | |
| | 5. BioNTech/Pfizer | | | | |
| | 195 (11.7) | 54 (11.8%) | 141 (11.7%) | | |
| | 6. I do not remember the name of the vaccine | | | | |
| | 246 (14.7) | 70 (15.3%) | 176 (14.5%) | | |

*Explanations regarding this value are given in Table 2.

COVID-19 after being vaccinated; Pfizer, with 141/195 individuals (72.3%); unknown vaccine—176/246 (71.5%); QazVac—211/305 (69.2%); CoronaVac—41/64 (64.1%); and Hayat-Vax—69/96 (60.9%).

No significant differences have been observed regarding the overall performance across five vaccines available for Kazakhstan's population (p 0.27).

## Discussion

### Attitudes toward vaccination

In our study performed at the end of 2022 in a sample of 2,009 participants, the proportion of those vaccinated increased almost three times (83.1%) compared to 2021, when the share of vaccinated people in Kazakhstan was 29.2% [9]. The mandatory vaccination of selected population groups introduced nationwide and the employers' requirements that provided 20.1% of those vaccinated contributed significantly to restraining the pandemic. In general, the experience of fighting the pandemic has substantially shifted the commonly accepted conception of vaccination. Historically, only the therapeutic paradigm—safety and benefit-risk assessment—was considered. A public health paradigm that strengthened its position after the pandemic considers the population impact and encompasses measures of community benefits against various outcomes, not only etiologically defined clinical consequences [16].

Our analysis showed that individuals with a college education were more willing to receive a COVID-19 vaccine. They provided two-thirds of the vaccinated (Table 1, 66%, p 0.001) and those who positively treated vaccination (S3 Table, 64.8%, p 0.001). In a study by Macharia et al. [17], 62.4% of respondents expressed willingness to receive COVID-19 vaccines, and they had similar characteristics, i.e., higher education and satisfactory income. Contrary to expectations, in our study, the most significant contribution to the proportion of those vaccinated was made not by high-income people but by middle-income earners (66.4% vs. 14.6% of high-income respondents, p.0.047). Our data on this issue are, to a definite extent, in contrast with other researchers' findings suggesting that individuals with lower socio-economic status tend to be more opposed to vaccination [5]. The perceptions of vaccination in those with higher education levels are also generally contradictory. Many researchers have reported that highly educated people's attitudes towards vaccination vary significantly. On the one hand, education positively affected individuals' vaccination acceptance in Ecuador, France, Germany, India, and the US. In contrast, a higher level of education was correlated with lower acceptance in Canada, Spain, and the UK [5, 18]. Hak et al. reported that among the determinants of an entirely negative attitude toward vaccination was also a high education ([OR] 3.3, [95% CI]: 1.3–8.6) [19].

Gender played no role in our study, neither in the attitude to vaccination nor in the acceptance of vaccines. There were no significant differences between genders in terms of the vaccination status of participants. However, men appreciated the national QazVac vaccine more favorably than women (52.7% trusted in QazVac, while 19.5% did not, p 0.001, S5 Table). In the study by Lazarus et al., which assessed the associations of age, gender, and level of education with vaccine acceptance, with a random sample of 13,426 participants, several noteworthy trends emerged: women in France, Germany, Russia, and Sweden were significantly more likely to accept a vaccine than men in these countries. Older (≥50) people in Canada, Poland, France, Germany, Sweden, and the UK were significantly more favorably disposed to vaccination than younger respondents, but the reverse trend held in China [18]. In our sample, there was no difference between age groups across vaccination status, but the youngest people aged 18–29 comprised 61.2% of those who welcomed vaccination (p 0.001, S3 Table). In the meantime, the same age group was mainly negatively disposed to the QazVac vaccine (56.4%, p

0.004, S5 Table). In another study by Lazarus et al., individuals aged 25–54 and 55–64 were more likely to get vaccinated upon the request of their employers. At the same time, most older people intend to take the vaccine in the future. As to the gender issues, men were found to be less likely to receive the vaccine than women [20]. Also, according to a survey of over 3,000 Australians regarding their attitude towards COVID-19 vaccination, women from socio-economically disadvantaged areas expressed more uncertainty regarding immunization acceptance than men (p 0.006) [21]. Our study displayed that individuals with chronic diseases showed a high propensity to be vaccinated and positively disposed of vaccines (p 0.01, S3 Table), albeit they did not trust the domestic vaccine (p 0.071).

According to Edwards et al., respondents with higher income levels who trust the government and medical institutions were likelier to favor vaccination [22]. In our study, we did not set out to find a relationship between income level and the degree of trust in official sources of information about COVID-19. However, we have identified the key factors that push a person to get vaccinated. Trusting official sources of information and a certain amount of relevant information subjectively assessed by respondents as "sufficient" increases the odds of vaccination by 69.3%. The same is valid for trust in the domestic vaccine when these predictors increase trust odds by 70.1%. In case the amount of information escalates to "many" or "too many," interviewees subjectively assess it as excessive, and the chance of trust in the QazVac vaccine or favorable decision regarding vaccination decreases or turns insignificant (S2 and S6 Tables).

Researchers have established that the lack of confidence in the safety of vaccines is one of the most significant factors influencing the acceptance rate [23]. Of the psychological determinants, confidence in vaccine safety was reportedly associated with one's own vaccination and recommendation behavior [24, 25]. According to a study from Malaysia, vaccine hesitancy among parents was 11.6%. In a univariate analysis, vaccination hesitancy was associated with unemployed status, younger age of parents, and other factors [26]. Among our respondents, we revealed 16.9% of those who refused vaccination (*N* 340) and 86.2% of them treated vaccination indifferently or negatively (chi-sqr. 246.8, p<0.001, Table 1). These respondents of different ages were primarily nonworking and had low incomes. They feared possible complications after vaccination (27.4%) and expressed anti-vaccine sentiments, being against all vaccines and not believing in the vaccine's effectiveness (38.2%). The overall % of those who did not trust vaccines in our study was 30.8% (Table 1, item 12).

According to Huang et al., the main reasons for vaccination hesitancy were concerns about the safety of the COVID-19 vaccine and limited trust in the government [27]. Other researchers have also mentioned that ideology, which can result in the rejection of science due to conspiracist ideation and limited trust in government medical experts, is a factor that indirectly affects immunization propensity [10, 28]. Huang et al. listed the presence of local vaccine distribution points among the factors contributing to vaccination [27]. In our study, respondents did not experience difficulties in reaching the vaccination points but faced challenges of other nature when receiving vaccines. Likely, many respondents chose multiple answers when asked about preferences and trust in the definite vaccine type (item 12, Table 1), as most did not receive the vaccine they were willing to get. The matter of trust in a particular type of vaccine was not related to the type of vaccine received due to the uneven supply of vaccines to healthcare facilities. Interviewees could not always choose the vaccine independently and received only the available ones. In particular, Sputnik V arrived first, and supplies with this vaccine were regular, while the BioNTech/Pfizer vaccine arrived last. This circumstance explains why almost half of those vaccinated (45.7%) received Sputnik V. Eventually, our survey did not shed light on the population preferences for five vaccines that have run in the country during the pandemic. However, it appeared reasonably

informative regarding trust in our national vaccine, QazVac (37.7% of respondents), and the general confidence in vaccines (69.2%).

## Barriers and acceptance of vaccination against COVID-19

El-Ghitany et al. have reported that the most common reasons for refusing the vaccine are concerns about its efficacy (39.5%) and possible side effects (38.8%) [29]. The top three concerns among the unvaccinated Black Americans were: (1) the vaccine is too new (92%), (2) safety issues (85%), and (3) preference for natural immunity over vaccine-induced immunity (85%) [30]. In a cross-sectional study from Saudi Arabia, the majority of participants received the Pfizer vaccine (72.8%) and most expressed high confidence in its effectiveness (55%) [31]. In our study, 195 out of 1,669 (11.7%) interviewees received the Pfizer vaccine, 54 fell ill after vaccination, and 141 (72.3%) avoided infection.

Fear of side effects, distrust of vaccination stakeholders, and a lack of confidence in the vaccine were barriers to COVID-19 vaccination among healthcare workers [32]. Researchers identified three major obstacles to vaccination: (1) lack of accessible and understandable information about a COVID-19 vaccine; (2) problems with access to vaccines and supplies; and (3) widespread misinformation based on mistrust of the medical community and government [33, 34]. Previously, we mentioned uneven supplies of vaccines that created certain obstacles to free choice in our respondents. The design of the questionnaire, which focused primarily on ease of survey and obtaining the essential information regarding attitudes toward vaccination, has not allowed for a more detailed study of the issues listed in the cited sources, particularly a level of mistrust of medical providers. Our analysis revealed that 14.5% of respondents relied on medical workers' vaccination recommendations.

A representative cross-sectional survey conducted in Germany with 27,509 participants found that older people and men were more likely to approve the compulsory vaccination than younger participants and women within the given context [35]. Other researchers stated the same situation [36]. There were no differences regarding gender in our research. At the same time, the proportion of retired people who disposed positively to vaccination was three times greater than those who treated immunization indifferently or negatively (18% vs. 6.1%, p 0.001, S3 Table).

Among 267 patients with inflammatory bowel disease in Poland, faith in vaccination was among the most common reasons for immunization (43.2%). The most common reasons for vaccine refusal were concern about side effects (50.0%) and possible exacerbation of gastroenterological disease (34.2%) [37]. More than two-thirds of our participants believed in vaccines (69.2%), and 212 of 268 (79.1%) respondents with chronic diseases got vaccinated. Our research suggests that not so much doubt about vaccine side effects plays a decisive role in the acceptance of vaccination and overall trust in vaccines, but rather, trust in different sources of information about COVID-19. In our study, two-thirds (66.5%) of all vaccinated respondents (1,669) trusted official sources of information about the disease. In comparison, 72.7% of those who refused (N 340) trusted alternative sources, i.e., YouTube podcasts, speeches of famous personalities, or opinions of their surroundings. The majority of vaccinated interviewees (75.2%) found the available official information "sufficient," while 57.7% of those who refused (340) considered the information "scarce" (p<0.001, Table 1). About three-quarters of those who positively treated vaccination, regardless of their vaccination status, found the official information "sufficient" (74.3%, p 0.001, S3 Table).

In this relation, the theory of planned behavior (TPB) developed by I. Ajzen applies to behavioral research on health issues [38]. According to the theory, "A more favorable attitude makes a person more attentive toward a recommendation made by significant others," i.e.,

family members, friends, relatives, and the most trustable media personalities. Our data are consistent with the findings of other researchers. They found that vaccine uptake and attitudes vary considerably between those who rely on different information sources to make their vaccination decisions [39]. Fridman et al. studied differential exposure to media channels and social networks in representatives of two American political parties. Researchers revealed that adherence to opposite information sources may explain these groups' divergence in perceived threat and vaccine attitudes [40]. The source of vaccine information is a crucial factor, and medical professionals play a unique role in improving knowledge and acceptance of vaccines among the public [41]. Hence, complete and reliable information from official sources and the medical community is necessary to overcome vaccine hesitancy.

## Advances and limitations of the study

We reported the explored population's attitudes toward vaccines in relevant detail, designated the reasons for refusal of immunization, and revealed the predictors of favorable vaccination decisions. We figured out the level of trust in the first national QazVac vaccine.

Regrettably, our study was characterized not so much by limitations as errors concerning the questionnaire's design and sampling. First was the substantial heterogeneity of our sample regarding age and gender. The youngest group of respondents exceeded the number of those of retirement age by more than four times, and the overwhelming majority of the sample were women. Such inequity significantly limited the chance of proper analysis across the groups, potentially leading to bias. An insufficient number of questions designed on a Likert scale can also be referred to a shortcoming. In addition, we failed to predict the pitfalls of composing some questions. In particular, we have not provided our tool with barriers for multiple responses to a single question. For instance, in the items regarding vaccine preferences and the section "The course of the disease in those who fell ill with COVID-19." These drawbacks substantially affected the questionnaire's statistical processing quality, reducing the range of possible findings to a definite extent.

## Conclusions

Within the explored sample of 2,009 individuals, most (83.1%) were immunized against COVID-19; among them, 20.1% obeyed the request of their employers that had been practiced in frames of non-pharmaceutical interventions to contain the disease. The youngest respondents, aged 18–29, individuals with a college education, students, employed people, and those with chronic diseases, showed a positive attitude toward vaccination (all $p < 0.05$). More than two-thirds of respondents (69.2%) expressed trust in all types of vaccines against COVID-19. Of those who refused vaccination (16.9%), about one-third feared vaccination consequences, and more than a third (38.2%) reported anti-vaccine sentiments.

The decisive factors in accepting vaccination were trust in official sources of information (reports of medical experts, etc.) and, mainly, subjectively interpreted sufficiency of information about the disease, which had increased the odds of being vaccinated by 63.9% (OR 1.71, 95% CI [1.3;2.26], $p < 0.05$).

Only 37.7% of respondents expressed confidence in the domestic QazVac vaccine, which may indicate insufficient educational efforts by public health structures in the country. History and severity of COVID-19 disease did not play a role in positive perceptions of vaccination, while illness after vaccination significantly affected vaccination approval (p 0.021). No significant differences have been observed regarding the overall performance across five vaccines (QazVac, Sputnik V, CoronaVac, Hayat-Vax, and BioNTech/Pfizer) available for Kazakhstan's population (p 0.27).

The vaccination experience during the COVID-19 pandemic should be thoroughly analyzed by the country's scientific research centers, health officials, and other involved structures.

## Supporting information

**S1 Table. The English version of the Questionnaire to evaluate the attitude toward vaccination and acceptance of the "QazVac" vaccine against COVID-19 in the West Kazakhstan population.**
(DOCX)

**S2 Table. Logistic regression analysis on relationships between the amount of information about COVID-19 vaccination in official sources and the vaccination status of respondents.**
(DOCX)

**S3 Table. Analysis of the relationship between attitudes toward vaccination and the other parameters, *N* 2,009.**
(DOCX)

**S4 Table. Analysis of the relationship between the history and severity of COVID-19 disease and trust in vaccines, *N* 2,009.**
(DOCX)

**S5 Table. Analysis of the relationship between trust in the "QazVac" vaccine and the other parameters, *N* 2,009.**
(DOCX)

**S6 Table. Logistic regression analysis on relationship between the amount of information about COVID-19 vaccination in official sources and the trust in the domestic "QazVac" vaccine.**
(DOCX)

**S7 Table. Analysis of the relationship between the disease after vaccination and attitudes toward vaccination, *N* 1,669.**
(DOCX)

## Acknowledgments

The authors' team thanks the University's Department of Biostatistics staff member Saltanat Zhumagaliyeva for assistance in the statistical processing of the results.

## Author Contributions

**Conceptualization:** Saltanat T. Urazayeva, Saulesh S. Kurmangaliyeva, Asset A. Kaliyev, Kymbat Sh. Tussupkaliyeva, Aisha B. Urazayeva, Zhuldyz K. Tashimova, Nadiar M. Mussin, Toleukhan Begalin, Aimeken A. Amanshiyeva, Gulaiym Zh. Nurmaganbetova, Shara M. Nurmukhamedova, Saule Balmagambetova.

**Data curation:** Saltanat T. Urazayeva, Saulesh S. Kurmangaliyeva, Asset A. Kaliyev, Kymbat Sh. Tussupkaliyeva, Aisha B. Urazayeva, Zhuldyz K. Tashimova, Nadiar M. Mussin, Toleukhan Begalin, Aimeken A. Amanshiyeva, Gulaiym Zh. Nurmaganbetova, Shara M. Nurmukhamedova, Saule Balmagambetova.

**Formal analysis:** Saltanat T. Urazayeva, Saulesh S. Kurmangaliyeva, Asset A. Kaliyev, Kymbat Sh. Tussupkaliyeva, Aisha B. Urazayeva, Zhuldyz K. Tashimova, Nadiar M. Mussin,

Toleukhan Begalin, Aimeken A. Amanshiyeva, Gulaiym Zh. Nurmaganbetova, Shara M. Nurmukhamedova, Saule Balmagambetova.

**Funding acquisition:** Saltanat T. Urazayeva, Saulesh S. Kurmangaliyeva, Asset A. Kaliyev, Kymbat Sh. Tussupkaliyeva, Aisha B. Urazayeva, Zhuldyz K. Tashimova, Nadiar M. Mussin, Toleukhan Begalin, Aimeken A. Amanshiyeva, Gulaiym Zh. Nurmaganbetova, Shara M. Nurmukhamedova, Saule Balmagambetova.

**Investigation:** Saltanat T. Urazayeva.

**Methodology:** Saltanat T. Urazayeva.

**Software:** Arman Issimov.

**Writing – original draft:** Saltanat T. Urazayeva.

**Writing – review & editing:** Saltanat T. Urazayeva, Arman Issimov.

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
