## [Decision Letter · Decision Letter 0]

10 Jan 2024

PONE-D-23-30867Attitude and acceptance of “QazVac” vaccine against COVID-19 among Aktobe city population, West Kazakhstan: a cross-sectional studyPLOS ONE

Dear Dr. Issimov,

Thank you for submitting your manuscript to PLOS ONE. After careful consideration, we feel that it has merit but does not fully meet PLOS ONE’s publication criteria as it currently stands. Therefore, we invite you to submit a revised version of the manuscript that addresses the points raised during the review process.

Please address referee #2 comment, and specify better the source of data and where are those checkable. Also, Ethics Committee clearance should be indicated, as per Plos One policies. 

We look forward to receiving your revised manuscript.

Kind regards,

Vincenzo Alfano

Academic Editor

PLOS ONE

Journal Requirements:

"This research was funded by the Science Committee of the Ministry of Science and Higher Education of the Republic of Kazakhstan (Grant No. AP14870878)."

Reviewers' comments:

Reviewer's Responses to Questions

**Comments to the Author**

1. Is the manuscript technically sound, and do the data support the conclusions?

Reviewer #1: No

Reviewer #2: Partly

2. Has the statistical analysis been performed appropriately and rigorously? 

Reviewer #1: No

Reviewer #2: Yes

3. Have the authors made all data underlying the findings in their manuscript fully available?

Reviewer #1: No

Reviewer #2: Yes

4. Is the manuscript presented in an intelligible fashion and written in standard English?

Reviewer #1: No

Reviewer #2: Yes

5. Review Comments to the Author

Reviewer #1: Dear Authors,

I must appreciate the efforts of the authors in conducting this research. However, I have lots of concerns regarding the manuscript. Overall, the manuscript Sections appear disconnected, and there is a need for a more systematic organization of ideas.

Here are a few major concerns that need consideration.

1. The manuscript contains numerous grammatical errors that significantly hinder the readability and overall quality of the writing.

2. The title of the manuscript does not seem to precisely reflect the content of the paper.

3. Moreover, the introduction lacks alignment with the subsequent sections of the manuscript specifically the introduction, result and conclusion of the manuscript. The structure of the introduction should be addressed differently, with the aim to make it logical and easy to read.

4. Regarding the methodology section, I suggest the authors add the Google questionnaire link, elaborate on the inclusion and exclusion criteria, and describe the income level and education level in the methodology section.

5. In my opinion I suggest authors consider adding WhatsApp group names as the majority of respondents were students.

6. In methodology ambiguous terminology should be avoided like focus group and etc.

7. In the questionnaire, the type of activity, education level, and family income level should be elaborated and explained. I suggest authors replace the presence of chronic disease with any medical history.

8. In attitude and trust in vaccination against COVID-19 few questions are not clear and vague like: What is your attitude to vaccination against COVID-19? What is your level of confidence in Kazakhstan's QazVac vaccine? Are these open-ended or closed-ended questions? Another question from the same section " Which COVID-19 vaccine do you trust" does not match with the title of the manuscript.

9. Question about the experience of COVID-19 is ambiguous and not clear like: Have you been ill with COVID-19 disease after vaccination? I suggest the authors modify this part of the question as it raises lots of concerns: After which dose of the COVID-19 vaccine do the respondents become ill? As few presents with symptoms after the first dose whereas other reports after 2nd dose. Another question which of the following symptoms did you have after the disease? I suggest the authors mention the symptoms, and the authors want to ask about symptoms after the disease or symptoms after COVID-19 vaccination???

10. I suggest authors use clear statements in the questionnaire and also mention the scale like the Likert scale or another scale that they used in their research.

11. The authors mentioned that “The oral informed consent of each respondent was obtained before filling in the questionnaire” How did authors collect this informed consent as this is an online questionnaire?

12. It is also mentioned in the methodology section that “All respondents were informed of the study purpose and procedure”. Do authors mention this in their online questionnaire???

13. Regarding, the result section I observed ambiguity in this section. The authors must provide clear and concise explanations of their findings, avoiding vague language or undefined terms. In Table 1 the age groups are not equally distributed; I recommend authors to distribute age groups equally. Table no: 4 and 5 is not well-explained, and their relevance to the study is unclear. I suggest the authors align the results with the title and rationale/objective stated in the introduction.

14. The total number of responses shown in Table 5 is much more than the study's sample size. Please recalculate the findings carefully.

15. The discussion section needs to be improved and should be rewritten with logical reasoning and a comparison of present study findings with other published studies. The reasoning presented in the discussion is unaligned with the findings and needs a cohesive link to the existing literature. I recommend the authors to focus the discussion on directly addressing the study's objectives. Remove any content that does not come up with the main argument.

16.The conclusion should be drawn on actual findings after the improvement of results and discussion and should be in line with the title and objective of the study. Authors are advised not to overestimate the generalization of present study results on an international scale.

I think that managing these issues will significantly improve the manuscript's quality and its possibility for publication.

Reviewer #2: Referee report

MS Number: PONE-D-23-30867

Title: Attitude and acceptance of “QazVac” vaccine against COVID-19 among Aktobe city population, West Kazakhstan: a cross-sectional study

Decision: accepted with moderate-major revisions

Short summary

The paper investigates the attitudes towards vaccination against COVID-19 and the level of confidence in the domestic QazVac vaccine in Aktobe city, West Kazakhstan, by means of a cross-sectional study. The results highlight some individual characteristics more correlated with the choice to get vaccinated. This is an important and emerging topic and it is focused in a area less covered by scholars. In my opinion these are good points to consider the paper to be published. Nevertheless, the manuscript needs some revisions in order to be suitable for POEN journal. In the following lines I will express my main concerns on the paper. I hope that my comments can be helpful for the author(s).

Recommended suggestions

1. In the abstracts (in the method subsection) I would suggest to add more information on the quantitative methodology used in the paper

2. The organization of the paper (in terms of sections sequence) is a little bit strange and unusual for me. The authors highlight the main findings of previous studies in section 4. I think that a background section after the brief introduction is needed (before data, methodology and results). In the results section the authors can make comments on the consistence between their results and previous results.

3. In the introduction section the authors should also briefly highlight that vaccination is one of the main policies aimed at reduce the risk of contagions. In fact, this is a pharmaceutical measure that has been supported also by non-pharmaceutical interventions (NPI). Please note that there are also some studies specifically focused on central Asia. I suggest some paper that could be probably useful in order to better frame this policy intervention among the public health policies chosen by national governments.

• Alfano, V. (2022). COVID-19 in Central Asia: exploring the relationship between governance and non-pharmaceutical intervention. Health Policy and Planning, 37(8), 952-962.

• Alfano, V., Ercolano, S., & Pinto, M. (2023). Modeling Central Asia’s Choices in Containing COVID-19: A Multivariate Study. Administration & Society, 55(9), 1819-1836.

• Perra, N. (2021). Non-pharmaceutical interventions during the COVID-19 pandemic: A review. Physics Reports, 913, 1-52.

• Mendez-Brito, A., El Bcheraoui, C., & Pozo-Martin, F. (2021). Systematic review of empirical studies comparing the effectiveness of non-pharmaceutical interventions against COVID-19. Journal of Infection, 83(3), 281-293.

4. The factors able to influence the choice to get vaccinated need to be better discussed. There are several papers with good literature review. I suggest the following papers just as starting point to be discussed in the background section

• Alfano, V., & Ercolano, S. (2022). Your vaccine attitude determines your altitude. What are the determinants of attitudes toward vaccination?. Vaccine, 40(48), 6987-6997.

• Baumgaertner B, Carlisle JE, Justwan F, Rabinowitz M. The influence of political ideology and trust on willingness to vaccinate. PLoS ONE 2018;13(1): e0191728.

• Czarnek G, Kossowska M, Szwed P. Political ideology and attitudes toward vaccination: A study report. Working paper; 2020

• Fridman A, Gershon R, Gneezy A, Capraro V. COVID-19 and vaccine hesitancy: A longitudinal study. PLoS ONE 2021;16(4):e0250123.

• Gessner BD, Kaslow D, Louis J, Neuzil K, O’Brien KL, Picot V, et al. Estimating the full public health value of vaccination. Vaccine 2017;35(46):6255–63.

• Hak E, Schönbeck Y, De Melker H, Van Essen GA, Sanders EA. Negative attitude of highly educated parents and health care workers towards future vaccinations in the Dutch childhood vaccination program. Vaccine 2005;23 (24):3103–7.

• Lazarus JV, Wyka K, Rauh L, Rabin K, Ratzan S, Gostin LO, et al. Hesitant or not? The association of age, gender, and education with potential acceptance of a COVID-19 vaccine: a country-level analysis. J Health Commun 2020;25 (10):799–807.

• Lewandowsky S, Gignac GE, Oberauer K, Denson T. The role of conspiracist ideation and worldviews in predicting rejection of science. PLoS ONE 2013;8 (10):e75637.

• Maurer J, Uscher-Pines L, Harris KM. Perceived seriousness of seasonal and A (H1N1) influenzas, attitudes toward vaccination, and vaccine uptake among US adults: does the source of information matter? Prev Med 2010;51 (2):185–7.

5. On the basis of the suggested literature, I think that the authors can improve the discussion and the interpretation of their results.

6. The paper is focused on the vaccine hesitancy that represent an important issue correlated with the aim of the manuscript. Nevertheless, in my opinion the authors should also consider other approach more based on the attitudes toward vaccination (that is probably more focused on their data). In fact, as pointed out by Alfano and Ercolano, the attitudes toward vaccination could represent a more complex and complete rationale rather than the simple hesitancy. In fact, if we measure individual attitude we can range from hesitancy to positive attitude toward vaccination.

7. The sample used in the paper is a convenience sample. I think that the authors should discuss this point in the paper. Moreover it could be interesting to have some statistics in order to compare some caracteristics of the sample population with the whole population of Aktobe (age, gender, education).

8. Honestly, I am not fully convinced of the quantitative methodology used by authors. In my opinion, in order to have a good ceteris paribus correlations the authors should use a regression model as the related literature. This kind of methodology could help the authors to go ahead the definition of the social profile of the residents of Aktobe who has been vaccinated. This is a very important point in order to improve the robustness of the paper. In the current version we know just some bivariate correlations but if I understood the methodology, the ceteris paribus condition is not respected. The authors should try to address this concern.

9. I think that the authors should try to deepener the policy implication of their study.

Other minor suggestions (not compulsory)

10. I appreciated some technical tables in the supplemental material (like the questionaiires) but I think that the tables with the main results should be in the main text.

6. PLOS authors have the option to publish the peer review history of their article (what does this mean?). If published, this will include your full peer review and any attached files.

Reviewer #1: **Yes: **Sadia Minhas

Reviewer #2: No

---

## [Author Response · Author response to Decision Letter 0]

15 Apr 2024

Our original research article has undergone the major revisions required and is detailed in the cover letter.

---

## [Decision Letter · Decision Letter 1]

2 May 2024

Attitude toward vaccination against COVID-19 and acceptance of the national “QazVac” vaccine in the Aktobe city population, West Kazakhstan: a cross-sectional survey

PONE-D-23-30867R1

Dear Dr. Issimov,

We’re pleased to inform you that your manuscript has been judged scientifically suitable for publication and will be formally accepted for publication once it meets all outstanding technical requirements.

Kind regards,

Vincenzo Alfano

Academic Editor

PLOS ONE

Additional Editor Comments (optional):

Reviewers' comments:

Reviewer's Responses to Questions

**Comments to the Author**

1. If the authors have adequately addressed your comments raised in a previous round of review and you feel that this manuscript is now acceptable for publication, you may indicate that here to bypass the “Comments to the Author” section, enter your conflict of interest statement in the “Confidential to Editor” section, and submit your "Accept" recommendation.

Reviewer #2: All comments have been addressed

2. Is the manuscript technically sound, and do the data support the conclusions?

Reviewer #2: Yes

3. Has the statistical analysis been performed appropriately and rigorously? 

Reviewer #2: Yes

4. Have the authors made all data underlying the findings in their manuscript fully available?

Reviewer #2: Yes

5. Is the manuscript presented in an intelligible fashion and written in standard English?

Reviewer #2: Yes

6. Review Comments to the Author

Reviewer #2: (No Response)

7. PLOS authors have the option to publish the peer review history of their article (what does this mean?). If published, this will include your full peer review and any attached files.

Reviewer #2: No
